# Systemic and Local Biocompatibility Assessment of Graphene Composite Dental Materials in Experimental Mandibular Bone Defect

**DOI:** 10.3390/ma13112511

**Published:** 2020-05-31

**Authors:** Alexandra Dreanca, Codruta Sarosi, Alina Elena Parvu, Mihai Blidaru, George Enacrachi, Robert Purdoiu, Andras Nagy, Bogdan Sevastre, Nechita Adrian Oros, Ioan Marcus, Marioara Moldovan

**Affiliations:** 1Pathophysiology/Toxicology Department, Faculty of Veterinary Medicine, University of Agricultural Sciencies and Veterinary Medicine, 3-5 Manastur Street, 400372 Cluj-Napoca, Romania; alexandradreanca@gmail.com (A.D.); genacrachi@gmail.com (G.E.); robert.purdoiu@usamvcluj.ro (R.P.); nagyandras26@gmail.com (A.N.); bogdan.sevastre@usamvcluj.ro (B.S.); oronadrian@yahoo.com (N.A.O.); ioan.marcus@usamvcluj.ro (I.M.); 2Department of Polymer Composites, Babes-Bolyai University, Institute of Chemistry Raluca Ripan, 30 Fantanele Street, 400294 Cluj-Napoca, Romania; mmarioara2004@yahoo.com; 3Pathophysiology Department, Faculty of Medicine, “Iuliu Haţieganu” University of Medicine and Pharmacy, 3/4 Victor Babeș Street, 400012 Cluj-Napoca, Romania; parvualinaelena@yahoo.com (A.E.P.); mihaidoc1@yahoo.com (M.B.)

**Keywords:** graphene oxide, dental composites, mandible defect, biocompatibility, osteo-inductive

## Abstract

The main objective of this research is to demonstrate the biocompatibility of two experimental graphene dental materials by in vitro and in vivo tests for applications in dentistry. The novel graphene dental materials, including one restorative composite and one dental cement, were subjected to cytotoxicity and implantation tests by using a rat model of a non-critical mandibular defect. In vitro cytotoxicity induced by materials on human dental follicle stem cells (restorative composite) and dysplastic oral keratinocytes (dental cement) was investigated at 37 °C for 24 h. After in vivo implantation, at 7 weeks, bone samples were harvested and subjected to histological investigations. The plasma biochemistry, oxidative stress, and sub-chronic organ toxicity analysis were also performed. The resulting cytotoxicity tests confirm that the materials had no toxic effects against dental cells after 24 h. Following graphene dental materials implantation, the animals did not present any symptoms of acute toxicity or local inflammation. No alterations were detected in relative organ weights and in correlation with hepatic and renal histological findings. The materials’ lack of systemic organ toxicity was confirmed. The outcomes of our study provided further evidence on the graphene dental materials’ ability for bone regeneration and biocompatibility.

## 1. Introduction

Graphene and graphene oxide-based nanomaterials have garnered great interest in research because of their unique physicochemical properties. Some of the chemical properties include large surface area, functionality, better conductivity, and good biocompatibility. The 2D allotropic structure allows the use in various biological fields (biochemistry, pharmacology, tissue engineering, regenerative medicine, etc.). Therefore, it is necessary to understand their interaction with the biological systems, which may lead to potential local and systemic toxic effects [1,2,3].

Graphene and its derivatives are generating considerable interest in terms of composite material design and are believed to be the next potential material used in developing dental materials [4]. They can be functionalized and combined with ceramics, polymers, and metals. The next generation of bio-composites with graphene has generated specific interest for the use in dentistry with graphene oxide being responsible for improving the physical, chemical, and mechanical properties of the biomaterials [5]. Additionally, the combination between graphene and water-soluble polymers and ceramics can improve the bioactivity and promote cell differentiation [5]. Furthermore, graphene is generating considerable interest in terms of bone tissue engineering such as having the ability to promote osteogenic differentiation [6]. Previous work has only focused on graphene in vitro toxicity, stating this material is not cytotoxic [7,8,9], but it may elicit a Toll-like receptor-mediated inflammatory response [10]. In vitro compatibility does not always translate to in vivo compatibility due to the possibility of an immune or allergic response and complicated metabolic pathways (bioaccumulation, biotransformation, metabolization, etc.). Although this approach is interesting and numerous in vitro studies can attest the graphene materials biocompatibility and osteogenesis capability, there is still considerable controversy surrounding in vivo animal studies regarding graphene potential biological effects in bone regeneration and dentistry. The biological concerns of graphene usage in nanocomposites and dental materials, in terms of physical properties, degradation mechanism, in vitro activity, and in vitro biocompatibility, are reported in different studies from the literature [11,12,13,14,15]. 

Several institutions like the FDA (Food and Drug Administration), ANSI (American National Standards Institute), ADA (American Dental Association), ISO (International Organization for Standardization), and certain accessory bodies, such as NIOM AS (Nordic Institute of Dental Materials) and EU (European Union), diligently work on the materials to be tested. ISO 7405 is exclusively for evaluating dental materials and ISO 10993 evaluates both medical and dental materials [16]. According to the standards of these organizations, any biocompatible materials must be free of risks [2]. Dental materials come into direct contact with the connective tissues of the teeth, oral mucosa, pulp, and the periapical tissues. Due to long term contact, the materials should be biocompatible, which means that they should not cause allergic reactions, suffer degradation, or have any toxic effects [17]. Tissue and systemic reactions are assessed after intraosseous implantation trough clinical, laboratory, imagistic, and histological assessments. An ideal osseous grafting treatment should involve the use of a bone inductive material that would be reliable, biocompatible, long-lasting, and capable of restoring mandible continuity with minimal injury [18]. Animal experimental models represent a critical element within the development and preclinical evaluation of new materials, aimed at promoting tissue regeneration, as they provide evidence to the European Committee for Standardization that new products or materials can be safe and efficacious in an in vivo setting [19].

Dental materials, such as resin composites and cement, can harm teeth and the surrounding soft tissues, and lead to hypersensitivity or other symptoms when applied in clinical practice. Therefore, test methods that mimic in vivo conditions, such as implantation, are more clinically relevant. The potential usage of graphene oxide in the development of dental materials field lead us to prepare composite materials and cements with graphene and to investigate a deeper understanding of the ability of graphene to improve the biological properties.

The synthesis of the experimental graphene dental materials (restorative composite and cement) and the experimental protocol for evaluating the biocompatibility and toxicity of these materials formulated for dentistry represent the novelty of this study. This study aims to evaluate the in vitro cytotoxic effects by cell viability measurements, validate the in vivo compatibility of graphene dental materials, to further widen current knowledge of graphene applications in dentistry and bio-medicine. The research also aimed to assess the local bone healing process induced by direct contact with the dental materials, and the durability of the materials during the study. 

## 2. Materials and Methods 

### 2.1. Composite Dental Materials

The materials used in this study include a light-curing hybrid restorative composite (GZ2) and light-curing hybrid cement (LC1). The organic matrix has been prepared by using a mixture of monomers: Bis-GMA (2,2-bis[4-(2-hydroxy-3-methacryloxypropoxy)phenyl]-propane) (synthesized in our laboratory) [20], UDMA (urethane dimethacrylate) (Sigma-Aldrich GmbH, Steinheim, Germany), and TEGDMA (triethylene glycol dimethacrylate) (Sigma-Aldrich GmbH, Steinheim, Germany) as a diluted monomer. The inorganic filler of the materials consists of powder mixtures, with different particle sizes, silanizated with γ-methacriloyloxypropyl-trymethoxysilane (A-174, Aldrich, Steinheim, Germany). The materials GZ2 and LC1 are light-curing initiated, with camphorquinone (0.5%) relative to the liquid mixture/amine (1%) as the initiator/activator system with the polymerization process being achieved using a Woodpecker LED-B Curing Light lamp (Guilin Woodpecker Medical Instrument, Co.Ltd., Guilin, China) for 40 s. The composition of the materials is detailed in Table 1. 

#### Cell Culture Tests of Materials

The assays were performed on two cell lines. Human (Caucasian) dysplastic oral keratinocytes (DOK cell line, accession no. ECCAC 94122104) were purchased from Sigma-Aldrich (Sigma-Aldrich Chemicals Company, Heidelberg, Germany) and dental pulp stem cells obtained at the Institute of Oncology, Cluj-Napoca, Romania. The protocol of obtaining the stem cells is detailed in the paper published by Oltean et al. [14]. Cells were cultured in Dulbecco’s modified Eagle’s medium (DMEM, Merck & Co., Kenilworth, NJ, USA), supplemented with 2 mM L-glutamine, 10% FBS (Fetal Bovine Serum), 5 μg/mL hydrocortisone, antibiotics, antimycotics, and, respectively, 5% non-essential amino acids (NEA) and beta mercaptoethanol for the pulp cells. All reagents were purchased from Sigma-Aldrich (Sigma-Aldrich Chemicals Company, Heidelberg, Germany).
Extract Preparation

The samples of GZ1 and GZ2 materials, light cured for 40 s with LED lamp, were incubated on medium, at a concentration of 3 cm^2^/mL, by complying with the ISO 10993-12:2012 proceedings, at 37 °C for 24 h. The obtained extract was filtered sterile through a 20-nm syringe filter and then used immediately for the experiments.
Viability assay

Cell survival was assessed through the colorimetric measurement of formazan, a colored compound synthesized by viable cells, using CellTiter 96® AQueous Non-Radioactive Cell Proliferation Assay (Promega Corporation, Madison, WI, USA). The cells: DOK and dental pulp cells, cultivated at a density of 10^4^/wells in 96 wells plaques (Techno Plastic Products AG, Zurich, Switzerland) were accommodated for 24 h, and then exposed to extracts of the samples diluted in medium for 24 h at a dilution range of 0–64 x. Viability was measured colorimetric, using an ELISA plate reader (Tecan, Männedorf, Switzerland) at 540 nm. For the GZ2 sample, dental pulp fibroblasts were used, while, for the LC1 sample, the dysplastic oral keratinocytes (DOK) cell line was employed. All experiments were done in triplicate. Untreated cell cultures were used as controls. Results are presented as percentages of untreated controls.
Flow Cytometry Assay

Dental pulp fibroblasts seeded at a density of 10^4^/cm^2^ in Petri dishes were exposed for 24 h to GZ2 sample extract in a dilution of 0.5 in medium, and prepared as described above at extract preparation. Untreated cells were used as controls. Cells were then washed and stained with Annexin V-fluorescein isothiocyanate (FITC)/vital dye propidium iodide (PI) (BD Pharmingen Biosciences, San Jose, CA, USA). Differentiation among these cell populations was done by flow cytometric detection using a BD FACS Canto II flow cytometer (Becton Dickinson & Company, Franklin Lakes, NJ, USA) equipped with two lasers as excitation sources: blue (488 nm, air cooled, 20 mW solid state) and red (633 nm, 17 mW HeNe) [21]. The data was analyzed using the BD FACSDiva Software (Becton Dickinson, Franklin Lakes, NJ, USA). A total number of 10,000 events were recorded for each sample. The fluorescence spectrum of Annexin V and PI were detected using a 530/30 nm and a 575/26 nm band-pass (BP) filter, respectively [21].

### 2.2. Animals

The experiment was performed on 20 healthy adult female Wistar rats, 8 months old, weighing 250 g on average. Animals were housed in the establishment for breeding and use of laboratory animals from the Faculty of Veterinary Medicine Cluj-Napoca, in standard conditions, at a temperature of 22–23 °C, humidity 55%, and 12-h light/dark cycle. They had free access to standard rodent granular food (provided by the Cantacuzino Research Institute, Bucharest, Romania) and water ad libitum. All experimental procedures that involved the use of laboratory animals followed the European guidelines and rules 337, as established through the EU Directive 2010/63/EU, and the Romanian low 43/2014, concerning animal protection and welfare. The Research Ethics Committee of the University of Agricultural Sciences and Veterinary Medicine Cluj-Napoca approved the experimental protocol and the State Veterinary Authority Cluj (Authorization 52/30.03.2017) authorized the experiment.

### 2.3. Experimental Protocols

In order to investigate the materials presented in Table 1, a non-critical mandible defect model was induced [22,23]. The anesthesia protocol was performed with Xylazine (6 mg/b.w) and Ketamine (60 mg/b.w) (Bioveta S.A., Ivanovice Na Hane, Czech Republic) [24]. Following shaving and disinfection, an incision overlying and paralleling the right mandible was carried down through the subcutaneous tissues, and the border of the mandible being exposed. The defects were made using a 2-mm cutting dental burr at 1500 rpm [23,25]. Muscle and skin layers were then closed with a polydioxanone suture, as presented in Figure 1 [25].

Postoperatively, all animals received analgesia with subcutaneous injections of tramadol (12.5 mg/bw; KRKA, d.d., Novo mesto, Slovenia). The animals received Enrofloxacin (0.2 mg/mL; Putney Inc., Portland, OR, USA) in the drinking water for seven days to prevent postsurgical infection. The animals were randomly divided into four groups, as follows: group 1 (n = 5), represented the control group, group 2 (n = 5) had a sham-surgery, and group 3 (n = 5) had a defect that was treated with light-curing hybrid dental cement (~15 wt/area) (LC1). Group 4 (n = 5) included the mandible defect that was filled with a light-curing hybrid dental restorative composite (~15 wt/area) (GZ2). Graphene dental material pastes were applied inside of the mandible defect and were light cured for 40 s using LED lamp. The animals were kept under close monitoring on the entire length of the study by focusing on infection prevention and analgesic therapy. The animals were weighed weekly to assess the nutrition status and radiographic analysis (Pausch technologies, Erlangen, Germany, IAE SpA tubes, Cormano, Italy) was performed 28 days after surgery. At the end of the experiment, blood samples were collected in order to determine oxidative stress and biochemical parameters. The animals were humanly euthanatized after seven weeks under prolonged narcosis. This was followed by cervical dislocation. Subsequently, the liver, kidneys, and mandible bone were harvested. The abdominal organs were weighted and relative organ weight was obtained for each rat by dividing organ weight on body weight.

### 2.4. Biochemistry

Biochemical analysis was carried out using the UV-VIZ Screen master Touch spectrophotometry analyzer (Hospitex diagnostics, Firenze, Italy). Urea, creatinine, alanine aminotransferase (ALT), aspartate aminotransferase (AST), alkaline phosphatase (ALP), creatinine kinase (CK), Calcium (Ca), and Phosphorus (P) were measured.

### 2.5. Oxidative Stress Analysis

The oxidative stress parameters determined within this experiment were: total antioxidant reactivity or capacity (TAR), total oxidant status (TOS), oxidative stress index (OSI), Malondialdehyde (MDA), Nitric oxide (NO), and Thiols (SH). All the parameters were measured using UV-Vis Jasco spectrophotometry analyzer (Jasco V-630, Tokyo, Japan). TAR and TOS of serum samples were determined using an automated measurement method. This method is essentially the same as that developed by Erel O, 2005 [26,27]. The ratio of total peroxide to total antioxidant potential was calculated in order to determine the OSI, which is an indicator of the degree of oxidative stress [28]. The assessment of NO was performed by the Griess reaction, which was described by Miranda K et al. 2001 [29]. The lipid peroxidation (MDA) has been determinate by the thiobarbituric reaction procedure reported by Mitev et al. 2010 [30]. The assessment of serum’s thiols was performed through the adapted method described by Ellman [31].

### 2.6. Histological Analysis

Bone samples from the mandibular lesion site, liver, and kidney samples were havested for the histological examination. Hepatic samples from the left lateral lobe and right medial lobe were taken, according to the recommendations of the Societies of Toxicologic Pathology from Europe (ESTP), Great Britain (BSTP), Japan (JSTP), and North America (STP) [32]. Half of the right, respectively, and the left kidney were also harvested. For decalcification, after fixation, the mandibula of the animals were kept in a mix of 8% formic acid and 8% HCl for 24 h and embedded in paraffin. The samples were fixed in 10% buffered neutral formalin, and embedded in paraffin. The sections were made at 4 micrometers and the slides were stained by the Hematoxiline–Eosine (HE) method. The slides were examined under a BX51 Olympus microscope (Olympus America, Inc., Melville, NY, USA) and images have been taken with an Olympus UC 30 digital camera (Olympus America, Inc., Melville, NY, USA) and processed using Olympus basic stream software. All the tissues’ sections have been examined by an independent observer blinded to the experimental protocol.

### 2.7. Statistical Analysis

For the cell culture tests, the statistical difference between experimental materials and control groups were evaluated two-way, ANOVA, and Student’s *t*-Test, which was followed by the Bonferroni post-test. All the values in text and figures are expressed as mean ± standard deviation. The results were considered significant for p ≤ 0.05. Statistical package used for data analysis was Prism version 4.00 for Windows, GraphPad Software, San Diego, CA, USA.

For the animal tests, all data are reported as Mean ± SEM. The Gaussian distribution was checked by the D’Agostino and Pearson omnibus normality test. Data were analyzed by one-way analysis of variance (ANOVA), which was followed by the post hoc Dunnett’s range test and the two-way ANOVA. This was followed by the Bonferroni post-test. Statistical significance was considered at p < 0.05 [95% confidence interval]. Statistical values and figures were obtained using GraphPad Prism version 5.0 for Windows, GraphPad Software, San Diego, CA, USA.

## 3. Results

### 3.1. Cell Culture Tests

Figure 2 is presented as the cell viability and flow cytometry analysis of dental pulp stem cells exposed to restorative composite GZ2.

GZ2 had no impact on the cell viability of the dental pulp fibroblasts (Figure 2) in the used concentrations of the extract. The viability of the cells was maintained above 98% of the untreated control readings, even at the highest concentration (p = 0.0549, not significant). This indicates that the material was very well tolerated by the cells. To confirm these results, a flow cytometry analysis was also performed, using the 0.5 dilution of the extract. As seen in Figure 2, the cells showed a significantly lower apoptosis rate in the treated cells group compared to the control group (p = 0.028). Therefore, the GZ2 material had no toxic effects against the dental pulp fibroblasts. Two-way ANOVA showed no significant interaction between the treated and untreated cells. 

LC1 extract was tested on dysplastic oral keratinocytes in different concentrations. The cell viability was increased when low concentrations were used (0.125, 0.25), and then gradually diminished with the concentration. However, it was still maintained above 77% of the untreated control when the undiluted extract was applied (Figure 3). Since the cell viability was above the toxic limit (70% of the untreated control) and the decrease was not significant (p = 0.69, t-Test, no significant treatment interaction showed by two-way ANOVA), the LC1 material was rather well tolerated by the cells and showed no toxicity in the present experimental setting. 

### 3.2. Clinical Assessment

Following graphene dental material implantation, the rats did not present symptoms of acute toxicity or local inflammation. The ingestion of solid foods was restarted after 12 h post-surgery and the body mass of the rats remained unaffected (p > 0.05) (Table 2). The X-rays revealed that the dental materials, GZ2 and LC1, were present in the bone defect during the experimental period, without any signs of resorption or disintegration of materials (Figure 4).

There were no significant differences among the body mass of the operated groups, in comparison with the control, LC1-dental cement group, and GZ2-restorative dental composite group (Mean ± SD) (two way ANOVA’s test, Bonferroni post-test, n = 5).

### 3.3. Oxidative Stress, Biochemical Analysis, and Organ Weight Analysis 

The mandible defect in the sham surgery group induced a significant increase in nitric oxide marker (NO) value (70.2 ± 3.2) in comparison to the control group (40.5 ± 2.7) (p < 0.001), LC1 group (49.5 ± 7.1) (p ≤ 0.001), and GZ2 group (53.2 ± 2.3) (p < 0.001). A reduction in thiols (SH) values in surgical groups 2 (0.4 ± 0.06) (p < 0.001), 3 (0.3 ± 0.06) (p < 0.001), and 4 (0.2 ± 0.01) (p < 0.001) was found, by comparison with the reference group (0.7 ± 0.07). Further analysis showed that there has been a progressive deterioration regarding oxidative status in the sham surgery group due to elevated total oxidative status (TOS) (24.4 ± 1.4) (p < 0.05), oxygen saturation index (OSI) (22.6 ± 1.3) (p < 0.01), and lipid peroxidation marker (MDA) (4.3 ± 0.9) (p < 0.05) values, by comparison with the control (TOS 19.5 ± 1.9, OSI 17.9 ± 1.7, MDA 3.0 ± 0.2). The conclusion which emerges from the data comparison was that graphene dental materials’ implantation appeared to confer better protection against oxidative stress (Table 3).

The biochemical analysis did not reveal renal injury because no significant difference was detected between urea and creatinine values (Table 4). Furthermore, there were no modifications found in the relative kidney weight of the animals (Figure 5a). The liver transaminases, alanine aminotransferase (ALT) and aspartate aminotransferase (AST) values were within the normal range, but the AST levels found in the LC1 group (110.3 ± 28.4) (p < 0.05) are lower than the control (173.5 ± 28.7) (Table 4). The graphene dental materials treatment had no effect on the liver ratio assessment (Figure 5b). The correlation between serum values and organ ratio is noteworthy because kidney and liver sub-chronic toxicity was not confirmed. The alkaline phosphatase (ALP) activity was elevated in the sham-surgery group (357 ± 132.1) (p < 0.01), and the dental materials groups (302.3 ± 41.09; 303.9 ± 58.2) (p < 0.01) in comparison to the control (133.4 ± 35.3). In the current study, there were no signs of muscular injuries due to the normal creatine kinase (CK) serum concentration (Table 4).

Graphene dental materials treatment significantly increased the level of serum calcium (Ca) (LC1: 15.5 ± 0.4) (p < 0.001) (GZ2: 14.1 ± 2.3) (p < 0.05) compared to the control group (10.4 ± 0.5). Accompanied by the calcium increase, changes in phosphorus (P) concentrations were identified in all operated groups (LC1: 6 ± 0.2 (p < 0.05), GZ2: 5.7 ± 0.3 (p < 0.01) in comparison to the control (7.1 ± 0.5). The mildly decreased concentration of the serum phosphorus from the experimental groups is in the normal range of the species (Table 4).

### 3.4. Histological Assessment

Histological samples from the liver and kidneys were evaluated. Liver sections from groups 1 and 2 (sham-surgery group) showed normal morphology without any kind of histological alterations. Samples from group 3 showed signs of slight congestion of the sinusoids from the midzonal and central areas. Scattered foci of mononuclear inflammatory cells were also observed near the central veins and portal spaces. Samples from group 4 showed normal hepatic histology with a few foci of mononuclear inflammatory infiltrate being observed near some portal areas (Figure 6). Histological sections from the kidney showed normal morphology in all of the experimental groups even though aspects of slight congestion was observed in group 3 (Figure 7). This congestion can occur incidentally and is considered to be a spontaneous background lesion [33].

### 3.5. Histology of the Bone Defect

In the sham-surgery group, the mechanically-induced bone defect was filled with a thin layer, consisting of granulation tissue (in both plexiform and oriented layers) and woven bone. The two components are well separated and no significant inflammation or active bone regeneration is noticed. The callus presented an irregular surface and was thicker than in the normal bone (Figure 8). In animals from a group treated with LC1, the mechanically-induced bone defect is completely filed by a finely-granular, basophilic to amphiphilic, amorphous material (tested material), delimited by a thick rim of proliferative woven and lamellar bone. Multiple minute stalks of proliferating bone are admixed with the tested material without a clear separation between them. Both the inflammation and the granulation tissue are largely absent, and the callus has an irregular surface (Figure 9). In animals from the group treated with GZ2, the bone defect was completely filled by a heterogeneous mixture consisting of fibrous connective and granulation tissue, with some inflammatory cells (mainly mononuclear) and an amphiphilic, granular material interspersed between the collagen layers. The marginal area of the defect has a rim of active osteoblastic bone formation and osteoclastic remodeling. 

Remarkably, in one animal, in some areas, a granulomatous reaction surrounding the tested material, with the presence of macrophages and giant cells as well as neutrophils and mature fibrous tissue, was observed (Figure 10).

## 4. Discussion

The present study has investigated the in vitro cytotoxic effects by cell viability measurements on the human dental follicle stem cells and on dysplastic oral keratinocytes in different concentrations. Furthermore, the systemic biocompatibility of the graphene dental materials through organ sub chronic toxicity evaluation, oxidative stress assessment, and histological examination were tested in vivo. 

The use of in vitro tests offers the possibility of studying the material’s biological compatibility. The advantages and disadvantages of the degree of polymerization of dental composites are fairly well established [34,35]. The toxicity of dental composites and resin formulations are related to the quantity and type of residual monomers and the pendant groups remaining after photo-polymerization. The use of graphene reinforcement and nanofiller has significantly enhanced their load carrying capacity and their mechanical performance. In our studies, we investigated the physico-chemical, mechanical, and antibacterial properties of experimental dental composites with different types of graphene. In addition, we have demonstrated that, by introducing graphene, the mechanical properties of the resulting biomaterial are enhanced [34,35,36]. 

Wanget et al. [37] demonstrated that graphene oxide had dose-dependent toxicity on human fibroblasts. The highly toxic effect was observed at concentrations above 50 g/m. A proposed mechanism related to cellular death involves the penetration of graphene oxide through the membrane into the cytosol and also via vesicle uptake. Graphene then generates reactive oxygen species (ROS) in the cytosol. This causes cell structure alterations and induces changes in the metabolic activity [38,39]. Additionally, composite material polymerization can be influenced by numerous factors such as composition, color, and light intensity emitted by curing units and light-curing time as well as thickness of overlying restorative material. For composites used in dentistry, monomers can diffuse through the polymerized bonding agent and dentinal tubules to reach the pulp tissue. These resin components are cytotoxic and may cause local tissue irritation and pulpal inflammation. Costa et al. [40] suggested that the cytotoxicity of the dental materials is inversely proportional to the light-curing time. Evaluation of the cytotoxic effects of dental materials on cell cultures has shown that the direct cell/material contact simulates the in vivo condition effectively [41]. Therefore, in this study, we performed a rapid 24-h cytotoxicity test in which the curing protocol of the materials with a light-curing lamp for 40 s was the same as that of the in vivo experimental protocol. In contradiction to the earlier findings, our GZ2 composite had no toxic effects on the dental pulp fibroblasts and LC1 was well tolerated by the cells and showed no cytotoxicity, which may lead to the possible future use of graphene oxide as fillers in various dental composite materials. Some studies found that graphenes were non-toxic on human dermal fibroblasts even at high doses, which is correlated with the results of our study [39]. 

Our experimental graphene dental composites are meant to be used in dentistry with their aim being to recover bone function and mastication. A mandibular defect was used for the biological evaluation of the experimental dental restorative composite and cement. Clinically, the local biological effects of material implantation are minimal. During the seven-week period of the study, the animals did not show signs of inflammation, difficulties in prehension, or defective mastication of pellet food. This is in good agreement with previous studies conducted on mandible defects in which mandible bone defect implants have shown that the materials did not induce any clear macroscopic evidence of infection or rejection during the healing process [42,43]. 

Oxidative stress is caused by increased production of reactive oxygen species [ROS], or by a decreased defense of the body’s natural antioxidant mechanisms. The biologic tests of resin-based dental materials are associated with increased oxidative stress caused by ROS [42]. Sources of increased ROS in the oral cavity include bleaching agents, dental cements, implant alloys, and composite fillings [43,44]. As previously reported, the results obtained demonstrated that both the surgical procedure of the defect and the graphene dental materials induced systemic oxidative stress, which was suggested by elevated levels of NO and low SH levels in all experimental groups. Our findings are in line with previous results, which have demonstrated that the combinations obtained with materials and graphene induced a high level of cellular oxidative stress by generation of mitochondrial ROS and cell death in mouse alveolar macrophages [43]. Despite these findings, the restorative composite did not decrease cellular proliferation and differentiation. High values of the total oxidant status markers and lipid peroxidation markers have been noticed only within the sham surgery group. According to our results, although oxidative performance was not ideal, the dental composites used in our study were able to enhance tissue healing and prevent greater oxidative damage caused by the surgical procedure. There are several possible explanations for these outcomes either being a consequence of the experimental surgery, handling, social behavior, or the relatively short period of our study [43]. Graphene oxide-based materials allow a wide range of biological outcomes and interpretation. 

Toxicological assessments are absolutely necessary when new composite dental materials are developed [45,46,47]. Therefore, the present study evaluated the sub chronic hepatic and renal toxicity through organ specific isoenzyme interpretation, organ ratio determination, and histopathology. No alterations in the relative organ weights were noticed and the tissue-specific enzymes were within the normal range of the species [48,49]. Additionally, hepatic and renal histological findings confirm the graphene dental materials’ lack of systemic organ toxicity. The correlation between the lack of in vitro cell toxicity and in vivo organ toxicity is worth noting because carbon-based materials present different effects when administered in vivo as they present different patterns of bio distribution [50]. Our experiment was found to be consistent with a previous study by Singh V. et al., 2019 [51], where a different carbon-based material was tested for toxicological effects by administering the material intravenously and measuring many of the same parameters in our study. The results put forward by Singh V. et al. [51], in terms of body weight fluctuation, biochemistry, hematology, oxidative stress, liver, kidney, and spleen toxicity, are similar to our findings and showed lack of general acute systemic and organ toxicity. Furthermore, our study demonstrated that these graphene dental materials did not contain toxic, leachable, or diffusible substances that can be absorbed into the circulatory system, which causes systemic responses. One of the most interesting results is that the restorative composite material promoted and enhanced alkaline phosphates and calcium activity. This concurs well with previous findings [52,53]. Alkaline phosphatase (ALP) is known to be an early marker for the osteoblastic differentiation [54]. Additionally, calcium is biologically active in the bone formation process, neuromuscular activity, cellular biochemical events, and blood coagulation [55]. The biochemical analysis did not reveal any significant differences regarding creatine kinase (CK) activity. The phosphorus (P) concentrations are decreased in all surgical groups due to the correlation between calcium and phosphorus serum values. Abnormal serum phosphorus concentration could have been caused by hormonal imbalances that affect serum calcium concentration [56,57].

Local toxicity is due to the chemical interaction of a toxic substance with biologically relevant molecules while tissue compatibility may also be dependent on causes other than material toxicity [58]. ISO 10993-6: 2007 regulates the testing method for in bone implantation. This test method is used for assessing the biological response of the bone tissue to an implanted material. The histological analysis revealed that both graphene dental materials were bone inductive. In group 3, the dental cement filled the surgical cavities without the development of inflammatory reactions. In the margins of the defect, a layer of bone was detected, which suggested a bone healing process. Our results corroborate with previous reports, showing the biocompatibility of an ionomer dental cement through osseous implantation [59,60]. In animals treated with graphene reconstruction composites, connective and granulation tissue were observed. Contrary to our expectations, an active osteoblastic bone formation was also noticed, which suggested the material is also bone inductive. Simultaneous to bone neoformation in the implant surface, the reconstructive material is phagocytosed by macrophages and multinucleated giant cells, which adds an inflammatory reaction to its properties. 

Further studies are required to elucidate the apparent lack of correlation between the contradictory results, inflammatory response on the one side, and bone regeneration on the other side. We are aware that our research may have other limitations. The first being that the study was conducted over a single timeframe and the second being that specific bone regeneration markers (immunohistochemistry) could not be analyzed, which reveals the difficulty of collecting data for the full process of regeneration. It is our recommendation that future studies should be conducted over a chronic period of time by focusing solely on the bone regeneration process.

The most important outcomes of this experiment are the ability of the graphene dental materials to induce bone formation in vivo. Our experiment confirms previous results obtained by Wu C et al., 2015 [56], where graphene-based scaffolds led to an increased rate of in vivo new bone formation in comparison to β-tricalcium phosphate scaffolds. 

## 5. Conclusions

We have evaluated the compatibility of two different dental materials with graphene, a restorative composite, and a dental cement, following bone implantation. Certain dental materials can harm teeth and the surrounding soft tissues, which leads to adverse reactions when applied in practice. Therefore, testing methods that mimic in vivo conditions, such as implantation, are more clinically relevant. 

Despite the limitations encountered, the present study indicated that the materials tested may not cause notable cytotoxicity reactions. The tested graphene dental materials have good systemic and local biocompatibility, even though the restorative material elicited some oxidative stress and a slight inflammatory response. The severity of the reaction noticed is thought to be acceptable. Taken together, these findings are the first step toward enhancing our understanding of graphene utilization in dentistry. Graphene dental composites could stand as promising candidates for future dental or bone cements with applicability in bone tissue engineering and dentistry.

## Figures and Tables

**Figure 1 materials-13-02511-f001:**
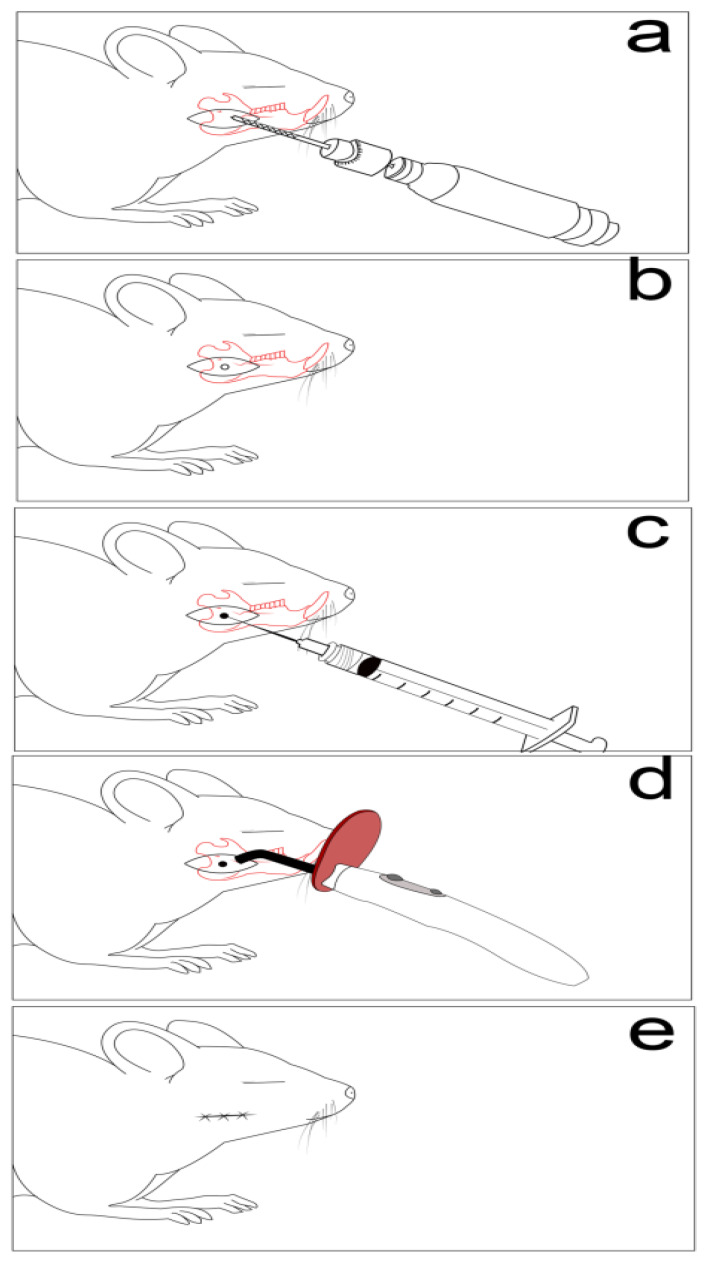
Non-critical mandible bone defect used in the experimental animals. (**a**) Defect drilling, (**b**) a circular bone defect (2/2 mm), (**c**) defect filling with graphene dental materials, (**d**) graphene dental materials polymerization, (**e**) and layers suture.

**Figure 2 materials-13-02511-f002:**
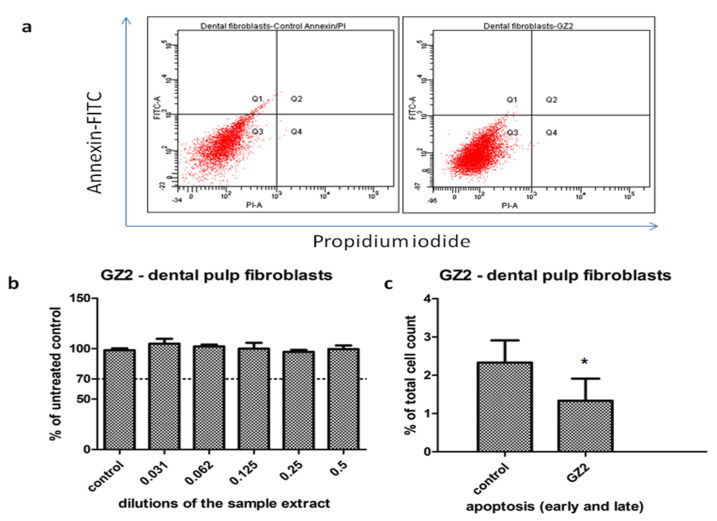
Cell viability and flow cytometry analysis of dental pulp stem cells exposed to GZ2 restorative composite. (**a**) Comparative flow cytometry diagrams of dental pulp stem cells untreated (left panel) and treated with GZ2, 0.5 extract dilution (right panel), following annexin-FITC/PI staining. The viable cells (showing no apoptosis) were identified as Annexin V (-)/ PI (-). The apoptotic cells were identified as Annexin V (+)/PI (-) (early apoptosis), and Annexin V (+)/PI (+) (late apoptosis). The differentiation among these three-cell-type populations was made by flow cytometric detection. (**b**) Graphical representation of the cell viability measurements of the dental pulp cells treated with different dilutions of the GZ2 extract. The results are expressed as the percentage of untreated controls, n = 3, media + SD, (**c**) Graphical representation of the apoptotic cells (early and late apoptosis), as measured by flowcytometry, n = 3, media + SD, t-Test was used for statistical analysis, *= p < 0.05.

**Figure 3 materials-13-02511-f003:**
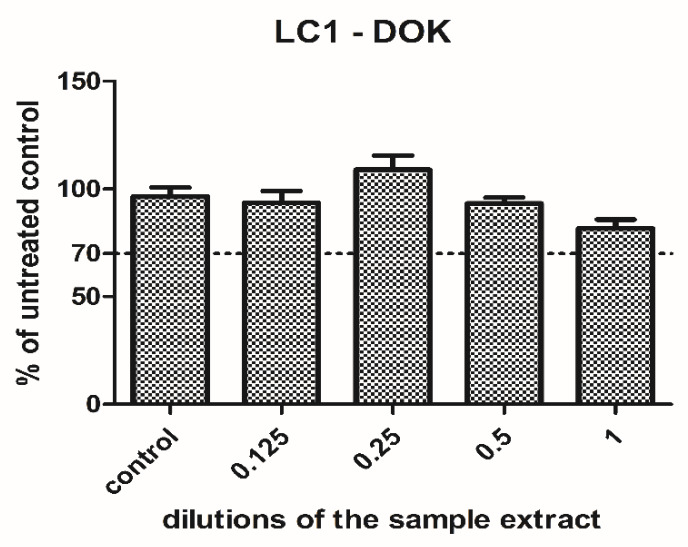
Cell viability of the dysplastic oral keratinocytes cells treated with LC1 extract in different dilutions. The results are expressed as the percentage of untreated controls, n = 3, media + SD.

**Figure 4 materials-13-02511-f004:**
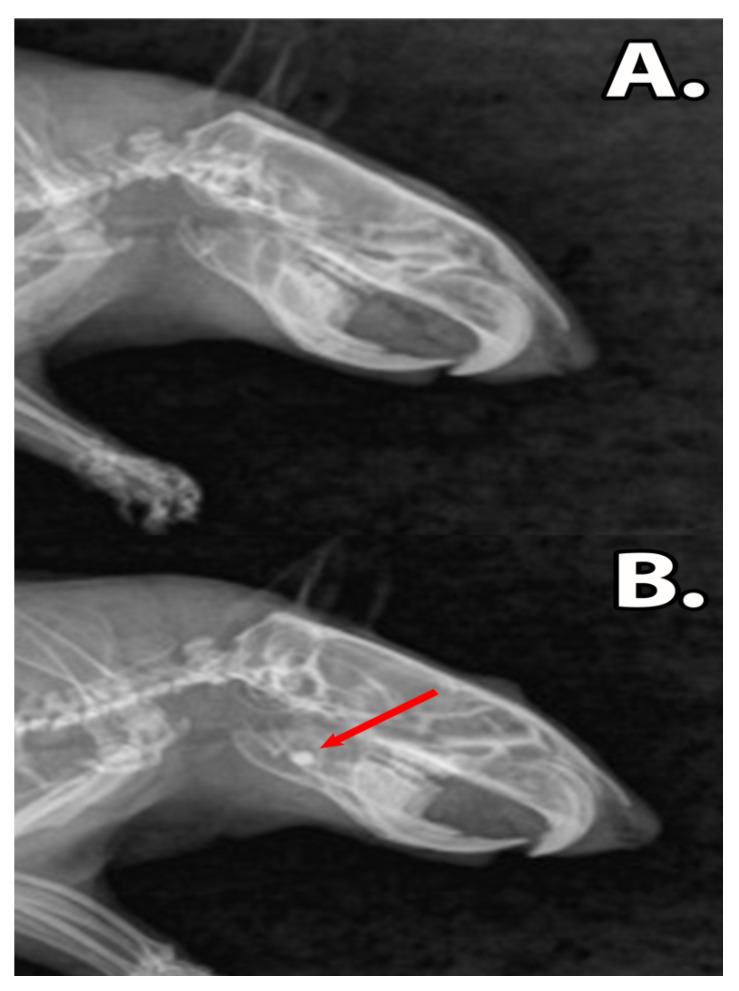
The X-ray images of the rat mandibles, (**A**) sham surgery group. The untreated defect is too small to be clearly observed. (**B**) Group filled with dental restoration material. The composites are clearly differentiated due to the radio opaque properties of the material (red arrow).

**Figure 5 materials-13-02511-f005:**
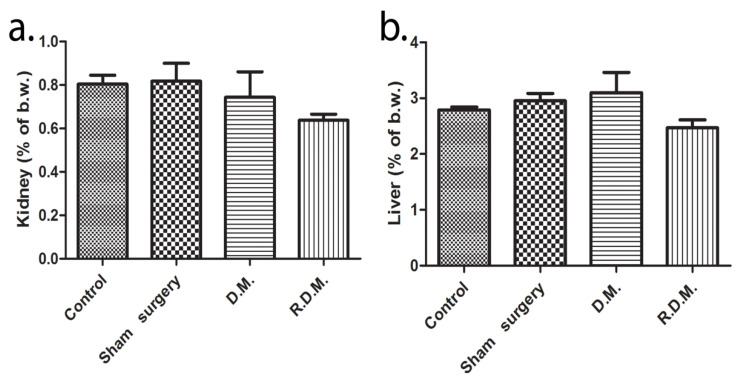
Changes in relative organ weight. (**a**) Kidney, (**b**) liver, (% of body weight), (one-way ANOVA test, post hoc Dunnett’s range test, n = 5).

**Figure 6 materials-13-02511-f006:**
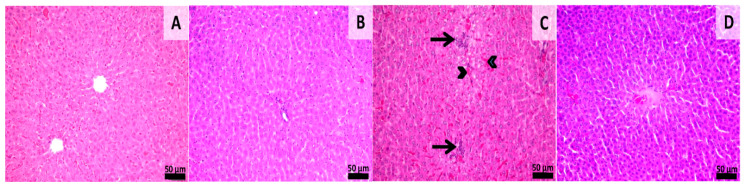
Histology of liver sections. Group 1 (**A**) shows normal hepatic morphology. Group 2 (**B**), normal morphology, Group 3 (**C**), Congestion of the sinusoids (arrowheads) in the mandibular and central areas, and mononuclear inflammatory infiltrate near the central veins and portal spaces (arrows). Group 4 (**D**), Normal hepatic morphology. HE stain, Scale bar = 50 µm.

**Figure 7 materials-13-02511-f007:**
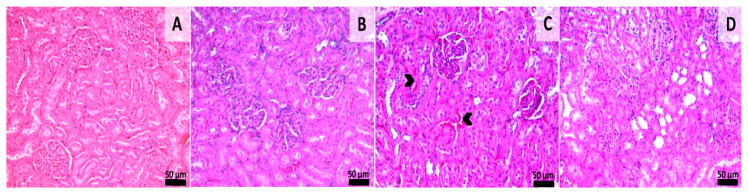
Histology of kidney sections. Group 1 (**A**) showing normal renal morphology. Group 2 (**B**), normal morphology Group 3 (**C**), Congestion in the renal cortex (arrowheads), Group 4 (**D**), normal renal morphology, HE stain, Scale bar = 50 µm. Normal renal morphology, HE stain, Scale bar = 50 µm.

**Figure 8 materials-13-02511-f008:**
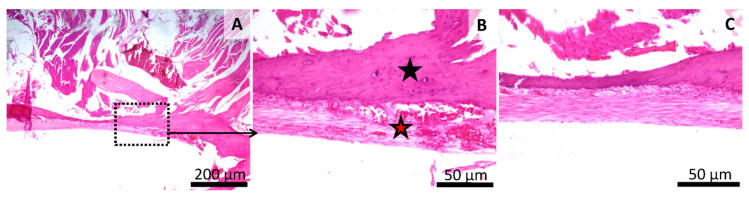
Histology of the mandibular defect. Sham-surgery group (Group 2), irregular aspect of the bone surface (black star) (**A**,**B**), thin layer of granulation tissue on the bone surface rich in blood vessels and fibroblasts (red star) (**A**,**B**,**C**). HE stain, Scale bar = 200 µm (**A**), Scale bar = 50 µm (**B**,**C**).

**Figure 9 materials-13-02511-f009:**
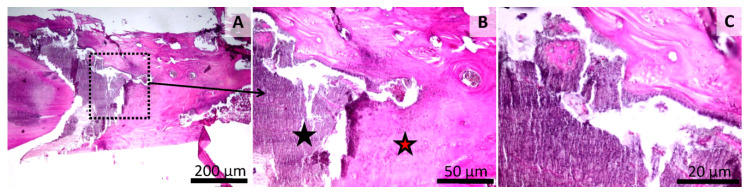
Histology of the mandibular defect. Dental cement Group (LC1) (Group 3), irregular aspect of the bone surface (**A**,**B**,**C**). Intramembranous and endochondral bone formation (red star) (**B**,**C**). Finely granular test material near the newly formed bone (black star) (**B**,**C**). HE stain, Scale bar = 200 µm (**A**), Scale bar = 50 µm (**B**), Scale bar = 20 µm (**C**).

**Figure 10 materials-13-02511-f010:**
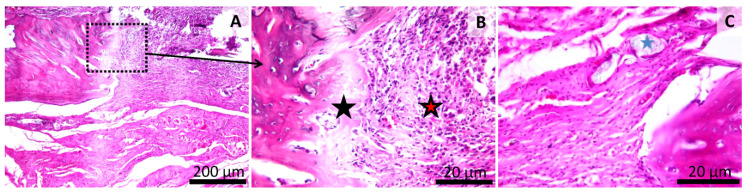
Histology of the mandibular defect. Dental restorative composite group (GZ2) (Group 4). Granulation tissue and fibrous tissue with mononuclear inflammatory cells (red star) (**A**). Area of osteoblastic bone formation (black star) (**B**). Fibrous reaction near the refringent test material (blue star). (**C**) HE stain. Scale bar = 200 µm (**A**), Scale bar = 20 µm (**B**,**C**).

**Table 1 materials-13-02511-t001:** The composition of the materials used within the experiment.

Materials	Type of Materials	Composition	Manufacturer
GZ2	light-curing hybrid restorative composite	Organic matrix: Bis-GMA, UDMA, TEGDMA. *Fillers*: 0.3% rGO/ZrO_2_; HA-ZrO_2_ (particle size 0.01–60 μm and 5–8 nm), silica, BaO glass (particle size 0.01–0.035 μm and 2–6 nm), quartz. 80% wt	UBB-ICCRR, Cluj-Napoca, Romania
LC1	light-curing hybrid cement	*Organic matrix*: Bis-GMA, UDMA, TEGDMA. *Fillers:* 0.3% TiO_2_-Ag-GO, silica, HA-F, BaF_2_ glass (particle size 2–6 nm). 70% wt	UBB-ICCRR, Cluj-Napoca, Romania

Bis-GMA-2,2-bis(3-(2′-hydroxy-3′methacryloyl-oxypropoxy)phenyl)propane, HA-ZrO_2_ hydroxyapatite–zirconia, HA-F fluorapatite, glasses (synthetized in UBB-ICCRR laboratory, Cluj-Napoca, Romania), TEGDMA-triethyleneglycol-dimethacrylate (Sigma-Aldrich GmbH, Steinheim, Germany), UDMA-urethane dimethacrylate (Sigma-Aldrich GmbH, Steinheim, Germany), DMAEM-2-dimethyl(aminoethyl)methacrylate (Sigma-Aldrich GmbH, Steinheim, Germany), Cq-camphorquinone (Aldrich, Steinheim, Germany), UBB-ICCRR, *Babes-Bolyai* University, Institute of Chemistry *Raluca Ripan*, Cluj-Napoca, Romania.

**Table 2 materials-13-02511-t002:** Post-surgery evolution of the body weight gain.

Weeks	Control Group	Sham Surgery	LC1 Group	GZ2 Group
1	253 ± 18.45	239.8 ± 23.95	242.8 ± 10.32	244.6 ± 28.58
2	275.4 ± 25.46	246.6 ± 25.61	248.8 ± 20.22	255.4 ± 26.21
3	283.8 ± 23.08	256.2 ± 27.9	270.2 ± 19.32	267 ± 17.94
4	297.6 ± 32.94	279.8 ± 31.09	284.3 ± 20.37	263.2 ± 23.79
5	308.4 ± 30.16	281.2 ± 32.61	292.8 ± 19.39	274.4 ± 15.54
6	314.8 ± 26.25	295 ± 32.82	297.8 ± 26.41	282.2 ± 4.32
7	326.2 ± 21.12	301 ± 30.74	313 ± 21.96	308.8 ± 10.77

**Table 3 materials-13-02511-t003:** Changes in the systemic oxidative stress status in response to the surgical bone defect and graphene material treatment.

	Control Group	Sham Surgery	LC1 Group	GZ2 Group
TAC	1.08 ± 0	1.08 ± 0	1.08 ± 0	1.08 ± 0
TOS	19.5 ± 1.91	24.4 ± 1.40 *	22.55 ± 1.17	22.46 ± 1.77
OSI	17.98 ± 1.75	22.68 ± 1.31 ***	20.98 ± 1.37 *	20.67 ± 1.63
NO	40.59 ± 2.74	70.27 ± 3.29 ***	49.50 ± 7.14 ***	53.23 ± 2.37 ***
MDA	3.05 ± 0.25	4.35 ± 0.95 *	3.42 ± 0.13	3.71 ± 0.92
SH	0.76 ± 0.07	0.43 ± 0.06 ***	0.36 ± 0.06 ***	0.27 ± 0.01 ***

Total antioxidant capacity (TAC), total oxidative status (TOS), OSI index, nitric oxide marker (NO), lipid peroxidation marker (MDA), thiols, (Mean ± SD) (one way ANOVA test, post hoc Dunnett’s range test, n = 5) * p < 0.05, p < 0,01, *** p < 0.001 in comparison to the control group. LC1-dental cement group. GZ2-restorative composite group.

**Table 4 materials-13-02511-t004:** Changes in biochemical parameters in response to the surgical bone defect and graphene dental materials treatment.

Parameters	Control Group	Sham Surgery	LC1 Group	GZ2 Group
Creatinine	0.46 ± 0.10	0.39 ± 0.03	0.41 ± 0.18	0.49 ± 0.19
Urea	18.96 ± 2.51	19.84 ± 3.93	19.68 ± 3.32	18.98 ± 4.20
ALT	43.14 ± 3.13	45.62 ± 3.17	40.24 ± 10.01	43.7 ± 9.46
AST	173.56 ± 28.73	153.92 ± 32.80	110.38 ± 28.48 **	129.7 ± 33.51
ALP	133.46 ± 35.3	357.08 ± 132.9 **	302.3 ± 41.09 **	303.96 ± 58.26 **
CK	227.8 ± 50.9	287.12 ± 134.1	208.78 ± 54.63	190.24 ± 41.28
Ca	10.42 ± 0.51	13.22 ± 2.19	15.52 ± 0.46 ***	14.18 ± 2.36 *
P	7.16 ± 0.58	6.72 ± 0.80	6 ± 0.29 *	5.74 ± 0.35 **

The variation of the kidney parameters (Urea, Creatinine), liver enzymes (Alanine Aminotransferase -ALT, Aspartate Aminotransferase-AST), Alkaline phosphatase (ALP), Creatine Kinase (CK), calcium (Ca), phosphorus (P) in the experimental groups. Physiological values: urea 13.2–27.1 mg/dl, creatinine 0.2–0.6 mg/dl, ALT: 16–48 U/l, AST: 65–203 U/l, ALP: 26–147 U/l, CK: 163–1085 U/l, Ca: 9.7–11.2 mg/dl, P: 5.58–10.7 mg/dl, (Mean ± SD) (one-way ANOVA test, post-test Dunnett, n = 5), * p < 0.05, ** p < 0.01, *** p < 0.001 in comparison to the control group. LC1-dental cement group, GZ2-restorative dental composite group.

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
