# Peer review of "Systemic and Local Biocompatibility Assessment of Graphene Composite Dental Materials in Experimental Mandibular Bone Defect"

_materials, 2020, doi:10.3390/ma13112511_

Round 1

Reviewer 1 Report

Comments to authors

Article “Systemic and local biocompatibility assessment of graphene composite dental materials in experimental mandibular bone defect” presents insightful in vitro gingival fibroblast and long-term in vivo exposure study in mice model. Article needs major revision to improve the quality of the paper. Most important English and grammatical errors are significant. At many occasion it is hard to follow the result due to poorly written draft. Major comments are appended below for the author’s perusal.

Please put the abbreviation of the major regulatory institution mentioned in introduction section like the Food and Drug Administration, American National Standards 44 Institute, American Dental Association, etc.

Page 2 , line 53-54 sentence ` Graphene oxide is believed to be the next potential material used in the development of dental materials` needs reference. Please cite the reports which give details of dental composites designed with graphene and allied materials with DOI: 10.2174/1381612822666151210124001

In experimental protocol, as described hybrid dental cement shown with wt/defect (~15 mg/defect), please modify the wt/area.

Authors need to put more details about FACS data analysis about how gating was performed to quantify the viability since that is a major evidence for the in vitro assay and very less described. Which kind of software was used, what criteria was used for y-x axis selection to interpret the data?

The articles as evident from title and introducing the content, tells about materials perspective in context with biocompatibility in vitro and in vivo, however, there is not material characterization. Authors need to add some physicochemical properties and mechanical properties to align the study design in context with biomaterials design for bone tissue integration as

Figure 4. changes in overall body weight into whole mice in control versus exposed groups is messy as individual are difficult to resolve, please modify the graph or chart type to make it comprehensive? This important to understand in clear way since Body weight changes are designated as important biomarkers for the in vivo toxicity.

In figure 9,10 and 11 there is no precise description about `C`, A and B are comprehendible that B is magnified (box) region. Please add these descriptions to figure legends to be clearly understood by readers.

Figure 6, 7 and 8 are not mentioned into the main text

In figure 7-8, can author mark the hallmark of liver and kidney visible in HPE micrograph to make it lucid for the readers and nonspecialist (e.g. Congestion in the renal cortex, and similarly other commented regions into manuscript)?

Figure 9-11, please mark/flag the different regions see in mandibular section to comprehend the anatomical features of sections.

Necrosis is more expected for such long term in vivo compatibility. In HPE, does authors notes any such abnormalities?

Can author demonstrate the ischemic zones with arrows in the HPE photograph shown?

Please statistically significant data values with * mark in all plots.

Do authors draw any time and dose dependent toxicity evaluation in different organs? Was impurity into graphene oxide, which often leads to toxicity into different organs reported in published literature, also noticed in this study?

Discussion line 401-404, sentence "  The correlation between the lack of in vitro cell toxicity and in vivo organ toxicity is worth noting, because carbon based materials present different effects when administered in vivo as they present different patterns of bio distribution [40, 41]..", authors need to cite recent reference related with ij vitro and in vivo seminal work on the carbon based biomaterials topic with https://doi.org/10.1039/c8tx00260f to update the reference list.

Minor

Typos, like ` imagistic `, ` in 96 wells plaques ` needs to be check throughout the draft

Author Response

Dear Reviewer,

The authors consider the reviewer’s comments and suggestions of highly scientific importance. Your ideas really helped us to improve the quality of our manuscript. We have corrected the errors that appeared in the manuscript and incorporated the changes in the revised manuscript. We hope the revised manuscript meets the expectations of journal standards and will be considered for publication soon.

Comments to authors

Article “Systemic and local biocompatibility assessment of graphene composite dental materials in experimental mandibular bone defect” presents insightful in vitro gingival fibroblast and long-term in vivo exposure study in mice model. Article needs major revision to improve the quality of the paper. Most important English and grammatical errors are significant. At many occasion it is hard to follow the result due to poorly written draft. Major comments are appended below for the author’s perusal.

Please put the abbreviation of the major regulatory institution mentioned in introduction section like the Food and Drug Administration, American National Standards 44 Institute, American Dental Association, etc.

Response: We added the abbreviation of the institution.

Page 2, line 53-54 sentence ` Graphene oxide is believed to be the next potential material used in the development of dental materials` needs reference. Please cite the reports which give details of dental composites designed with graphene and allied materials with DOI: 10.2174/1381612822666151210124001

Response: We added the recommended reference. It is number 6.

In experimental protocol, as described hybrid dental cement shown with wt/defect (~15 mg/defect), please modify the wt/area.

Response: We changed the wt/defect with wt/area.

Authors need to put more details about FACS data analysis about how gating was performed to quantify the viability since that is a major evidence for the in vitro assay and very less described. Which kind of software was used, what criteria was used for y-x axis selection to interpret the data?

Response: Data was added in the materials and methods section and the figure legend.

The articles as evident from title and introducing the content, tells about materials perspective in context with biocompatibility in vitro and in vivo, however, there is not material characterization. Authors need to add some physicochemical properties and mechanical properties to align the study design in context with biomaterials design for bone tissue integration as

Response: In this manuscript the physicochemical and mechanical properties of the investigated materials are not mentioned due to the following facts: the mechanical tests of composites with graphene were published in papers (References 34, 35, 36); the results demonstrated the improvement of mechanical properties at a very low filler content of graphene. We also have a manuscript in reviewing process that contains mechanical tests on GO-ZrO2 and GO-SiO2 entitled: Assessments of antibacterial and physico-chemical and mechanical properties for dental materials composites with graphene.

Figure 4. changes in overall body weight into whole mice in control versus exposed groups is messy as individual are difficult to resolve, please modify the graph or chart type to make it comprehensive? This important to understand in clear way since Body weight changes are designated as important biomarkers for the in vivo toxicity.

Response: We have modified the graph into a more comprehensive table that shows body weight fluctuations throughout the study.

In figure 9,10 and 11 there is no precise description about `C`, A and B are comprehendible that B is magnified (box) region. Please add these descriptions to figure legends to be clearly understood by readers.

Response: In figure 9, C image is half of the magnified area highlighted in A highlighting an area with more granulation tissue on the bone surface and fibroblasts, the other half being highlighted in B with the stars showing each area.

In figure 10, C image is a more magnified view of B, but it is the same area.

In figure 11, we have modified the images with stars, highlighting different areas.

Figure 6, 7 and 8 are not mentioned into the main text.

Response: The three figures are referred to in the text.

In figure 7-8, can author mark the hallmark of liver and kidney visible in HPE micrograph to make it lucid for the readers and nonspecialist (e.g. Congestion in the renal cortex, and similarly other commented regions into manuscript)?

Response: We have modified the images accordingly with arrows.

Figure 9-11, please mark/flag the different regions see in mandibular section to comprehend the anatomical features of sections.

Response: We have modified the images accordingly with stars.

Necrosis is more expected for such long term in vivo compatibility. In HPE, does authors notes any such abnormalities? Can author demonstrate the ischemic zones with arrows in the HPE photograph shown?

Response: Bearing in mind that our study was only 7 weeks in duration there were no areas of ischemia or necrosis observed. This duration is considered to be subacute, it would be possible to develop necrosis after a longer period of time in a chronic study (more than 3 months), according to ISO 10993- 6 standards.

Please statistically significant data values with * mark in all plots.

Response: If there was no statistical significance detected in a specific plot, then no star was added, otherwise stars were added wherever neccesary.

Do authors draw any time and dose dependent toxicity evaluation in different organs? Was impurity into graphene oxide, which often leads to toxicity into different organs reported in published literature, also noticed in this study?

Response: There was no organ toxicity found as we cannot consider congestion as being a hallmark of organ toxicity, according to Wallig et al, 2018 (Fundamentals of Toxicology Pathology). To the best of our knowledge the derived biomaterial is without impurities, this is deduced by the lack of cytotoxicity and organ toxicity.

Discussion line 401-404, sentence " The correlation between the lack of in vitro cell toxicity and in vivo organ toxicity is worth noting, because carbon based materials present different effects when administered in vivo as they present different patterns of bio distribution [40, 41]..", authors need to cite recent reference related with in vitro and in vivo seminal work on the carbon based biomaterials topic with https://doi.org/10.1039/c8tx00260f to update the reference list.

Response: The reference has been mentioned in text 427-434. The reference number is 52.

Reviewer 2 Report

Comments:

1.) Authors must describe novelty of the work.

2.) Introduction did not follow the logical progression of the subject. It should give entire description of the emerging problem and how the authors can address it through their study. However, it contains known background for the use of graphene oxide as dental material implantation regardless of comparing them with other NMs for example, or even showing the challenges arisen from traditional carbon materials.

3.) The English needs to improve and the grammar and spelling have to be checked carefully.

4.) Results and Discussion sections have to be potentially improved.

5.) Typos in Figure 2 presented as “Figure 2. Fig X” has to be edited.

5.) Many references do not conform with the journal guidelines (ex. 14, 20, 25, 32, 33, 35, 39).

Author Response

Dear Reviewer,

The authors consider the reviewer’s comments and suggestions of highly scientific importance. Your ideas really helped us to improve the quality of our manuscript. We have corrected the errors that appeared in the manuscript and incorporated the changes in the revised manuscript. We hope the revised manuscript meets the expectations of journal standards and will be considered for publication soon.

Comments:

  • Authors must describe novelty of the work.

Response:  We added the novelty of work.

  • Introduction did not follow the logical progression of the subject. It should give entire description of the emerging problem and how the authors can address it through their study. However, it contains known background for the use of graphene oxide as dental material implantation regardless of comparing them with other NMs for example, or even showing the challenges arisen from traditional carbon materials.

Response: We have modified.

  • The English needs to improve and the grammar and spelling have to be checked carefully.

Response: We improved the grammar and spelling.

  • Results and Discussion sections have to be potentially improved.

Response: We improved the Results and Discussion sections.

  • Typos in Figure 2 presented as “Figure 2. Fig X” has to be edited.

Response: We changed.

  • Many references do not conform with the journal guidelines (ex. 14, 20, 25, 32, 33, 35, 39).

Response: We corrected the form of the references, but because we introduced more references, the numbers has been changed. So, 14 become 18; 20 become 24; 25 became 29; 32 became 42; 33 became 43; 35 become 45; 39 became 49).

Reviewer 3 Report

The article is interesting and very promising. 

However there are some issues before manuscript can be accepted. 

Line 43: which biological fields?

line 50: conjunctive..do You mean connective tissues?

Line 53-55: resume in one sentence

Line 64-65: I can't understand. 

Methods 

the methods should match the abstract. In the abstract you indicated with the acronym of the materials the type of cells.correct please

Line 107: FCS..do you mean FBS?

Discussion: can you please discuss the wet environment influence on the curing material and the polimerization?

Author contribution should be assembled according to the guidelines. 

Author Response

Dear Reviewer,

The authors consider the reviewer’s comments and suggestions of highly scientific importance. Your ideas really helped us to improve the quality of our manuscript. We have corrected the errors that appeared in the manuscript and incorporated the changes in the revised manuscript. We hope the revised manuscript meets the expectations of journal standards and will be considered for publication soon.

Comments and Suggestions for Authors

The article is interesting and very promising. 

However there are some issues before manuscript can be accepted. 

Line 43: which biological fields?

Response: Yes, it has been elucidated

line 50: conjunctive..do You mean connective tissues?

Response: Yes, it has been changed.

Line 53-55: resume in one sentence.

Response: Yes, it has been changed.

Line 64-65: I can't understand. 

Response: Yes, it has been elucidated.

Methods 

the methods should match the abstract. In the abstract you indicated with the acronym of the materials the type of cells.correct please

 Response: We changed the abstract.

Line 107: FCS..do you mean FBS?

Response: Yes it is FBS. We modified FCS into FBS.

Discussion: can you please discuss the wet environment influence on the curing material and the polimerization?

Response: In our protocol, the results confirm that environment didn’t influence the polymerization of the materials. Previously, the degree of conversion and residual monomers of the materials (GZ2, LC1), were evaluated and the results were good, despite the presence of graphene in composition.   

Indeed, numerous factors may affect the polymerization of resin restorative materials, such as the composition, color, light intensity emitted by the curing units and the light-curing time, as well as thickness of overlying restorative material. For composite use in dentistry, monomers can diffuse through the polymerized bonding agent and dentinal tubules to reach the pulp tissue. These resin components are cytotoxic and may cause local tissue irritation and pulpal inflammation.

Author contribution should be assembled according to the guidelines. 

Response: We added the author contribution.

Many references do not conform with the journal guidelines (ex. 14, 20, 25, 32, 33, 35, 39).

Response: We corrected the form of the references, but because we introduced more references, the numbers has been changed. So, 14 become 18; 20 become 24; 25 became 29; 32 became 42; 33 became 43; 35 become 45; 39 became 49).

Reviewer 4 Report

The present study aims to evaluate through: i) in vitro cell viability experiments on two cell lines (dental pulp fibroblasts and dysplastic oral keratinocytes) the cytotoxic effects and ii) through in vivo analysis the compatibility of two novel graphene-based composites, to further deepen the current knowledge behind graphene applications in dentistry. Although it is almost known that graphene is a promising material in dentistry, the authors should highlight why this study. There are a lot of studies on this topic. Moreover, changes should be performed in the whole text. It is strongly recommended that the whole text must be revised by a native English speaker. Mistakes and typos are present in the whole text. Please revise carefully it. Specifically: 

-In the abstract section please specify the two kinds of novel graphene-based composites to be immediately comprehensive for the readers. The lines of cells are not specified too. Moreover, the abbreviations GZ2 and LC1 in the abstract are referred to the cell types used meanwhile in the materials and methods section they refer to the types of novel graphene based composite  materials.  The abstract should be reorganized and described in a more scientific way.

- introduction: there are no references at all in the first paragraph. Please insert. Moreover, reference 1 is mentioned different times in the introduction section. The whole section should be revised and better organized. In the introduction it is mentioned the graphene material, but no studies are related to the two novel types of graphene based composite materials of the present study. A careful search should be performed and highlighted in the introduction section the presence or not of these studies.

Moreover, different references are missing such as line 43

-Materials and methods section:

Line 84 2.1. Composite dental materials

This section should be better organized and described including all the materials details as they are missing in the text. In the present form it is very confusing. Table 1 is not presented in a scientific way it is very confusing. It is not comprehensive the lines 97 to 102 to which word they are related to. Please revise the whole section and specify the details of the materials e.g ( cell lines, it is not sufficient Heidelberg, Germany, please insert the missing information in the whole section of materials and methods. In addition, the authors should refer to the instructions for the authors to better organize not only the figure captions. Please better describe the later in the text.  (Ex. “Figure 7 Histology of liver sections. Control group (A), showing normal hepatic morphology; Sham  operated Group (B), normal morphology, Group 3 (C) Congestion of the sinusoids in the mandibular and central areas, mononuclear inflammatory infiltrate near the central veins and portal.” It is such confusing and seems that the authors have not read carefully the whole manuscript after writing it.  The same thing happens for figure 8. Please check all the figures. The authors  first refer to a “Control Group” and then to a “Group 3”, is not clear and uniform. This is not presented in scientific style.

Please specify why you used only 24 h as period for the cell viability and not different time point to better understand what happens. This kind of experiment is recommended, otherwise the results are skimpy. Moreover, there are not optical microscopy images which shows the viability of the two cells for each group.

-In the discussion section, line 371 “Our experimental graphene based composites are meant to be used in dentistry, being able to 371 recover bone function and mastication. Why the authors state such a strong definition. Did you analyzed the mechanical characteristics of these two materials to state their mastication (loading support) function? The whole section should be revised and better organized. It is confusing.

Line 387. Previously this section you state about your results now you say: “Despite these findings, the composite did not decrease cellular proliferation and differentiation and put a reference on it. Please correct this section too.

-Conclusions, it should be stated that the cytotoxicity was performed only at one time point (24h) it is so risky to state that these materials have no cytotoxicity effects. I think that this is a big limit of this study.

 Moreover, in the conclusions it is stated “The clinical compatibility and durability of the material is confirmed with signs of neither disintegration nor toxicity. ..” disintegration? What do you mean with it?  Then….”Taken together, these findings are the first step towards  enhancing our understanding of graphene utilization in dentistry.” You did not analyzed graphene materials but graphene based materials.

Please include also the limits of the present study.

Furthermore, the references (25,32,33,39,40,41,47) are not presented in the correct form. Please refer to the author journals guidelines for the references. You should put it in the right format with the year in bold.

Author Response

Dear Reviewer,

The authors consider the reviewer’s comments and suggestions of highly scientific importance. Your ideas really helped us to improve the quality of our manuscript. We have corrected the errors that appeared in the manuscript and incorporated the changes in the revised manuscript. We hope the revised manuscript meets the expectations of journal standards and will be considered for publication soon.

Round 2

Reviewer 1 Report

Authors made substantial changes to improve the quality and content of the paper. I am happy with the changes and response from the authors. The research article can be accepted now for the publications.

Author Response

Dear Reviewer, 

Thank you very much for your revision. We are happy that changes made are according to your expectation and the article can be accepted now for the publication.

Best regards,
Codruta Sarosi

Reviewer 2 Report

The authors addressed my questions properly. Therefore, I accept it in its current format.

Author Response

(The authors gave the same response as above.)

Reviewer 4 Report

Dear authors, 

your work has improved after the first review. However, there are still different issues to be corrected. Please all the changes you make should be highlighted through the word track changes record function. In this way it will be more easy and fast for the reviewers to review your paper. 

Abstract section: please insert the background. You started the abstract directly with materials and methods section. Although you should not report the single sections with subtitles in the abstract section you  should respect them to have a high quality paper. Moreover, the abstract must be changed as it is not yet fluent and has different flows. The same for the other sections. In the introduction section still miss information regarding the two materials they the authors studied. Still are missing different information on the materials and methods section. Please, see the instruction for the authors when you report a material, medium, cell line etc, which information should be included.

On the materials and methods section you reported: "Dental pulp stem cells were kindly donated by Dr. Olga Soritau, PhD from Institute of Oncology, Cluj-Napoca, Romania [14]." What do you mean with this sentences. It is not clear. Why the references? for which purposes? Indicating the procedure for collecting the stem cells? I am missing it.

The authors improved the discussion section but it still should be checked for flaws or errors. They also included the limits of the present study as the reviewer requested but they should shorten a little bit the conclusion section, as it is too long. However, being still messy it is difficult to understand what the authors performed in this study. 

Author Response

Dear Reviewer,

Thank you very much for your revision. We answered to all observation and we hope that now all the changes as according with expectations.

Dear authors,

Your work has improved after the first review. However, there are still different issues to be corrected. Please all the changes you make should be highlighted through the word track changes record function. In this way it will be more easy and fast for the reviewers to review your paper.

Abstract section: please insert the background. You started the abstract directly with materials and methods section. Although you should not report the single sections with subtitles in the abstract section you should respect them to have a high quality paper. Moreover, the abstract must be changed as it is not yet fluent and has different flows. The same for the other sections. In the introduction section still miss information regarding the two materials they the authors studied. Still are missing different information on the materials and methods section. Please, see the instruction for the authors when you report a material, medium, cell line etc, which information should be included.

Response: We reformulated the suggested points.

On the materials and methods section you reported: "Dental pulp stem cells were kindly donated by Dr. Olga Soritau, PhD from Institute of Oncology, Cluj-Napoca, Romania [14]." What do you mean with this sentences. It is not clear. Why the references? for which purposes? Indicating the procedure for collecting the stem cells? I am missing it.

Response: We changed.

The authors improved the discussion section but it still should be checked for flaws or errors. They also included the limits of the present study as the reviewer requested but they should shorten a little bit the conclusion section, as it is too long. However, being still messy it is difficult to understand what the authors performed in this study.

Response: We shorted the conclusion section and we revised all the manuscript. Track-changes was employed for manuscript correction.

Round 3

Reviewer 4 Report

Dear authors,

Your work has improved after the second review. However, there are still different issues to be corrected. Please check the bibliography again because there are still errors(33,50,51). Please revise the entire manuscript before publication.

Author Response

Dear reviewer,

Thank you for your observations and for the time given to review this manuscript.
We changed the recommended bibliography according to template.
We revised all the manuscript, and we used "Track Changes" function for better visualization of the modifications.

Best regards,

Codruta Sarosi